# Adaptive Reuse of a Historic Building by Introducing New Functions: A Scenario Evaluation Based on Participatory MCA Applied to a Former Carthusian Monastery in Tuscany, Italy

**Agnese Amato** [1] , **Maria Andreoli** [2] **and Massimo Rovai** [3,*]

1   Department of Energy, Systems, Territory and Construction Engineering, University of Pisa, 56100 Pisa, Italy; una.amato@gmail.com
2   Department of Agricultural, Food and Agri-Environmental Sciences (DAFE), University of Pisa, 56124 Pisa, Italy; maria.andreoli@unipi.it
3   Department of Civil and Industrial Engineering (DCIE), University of Pisa, 56126 Pisa, Italy
*   Correspondence: massimo.rovai@unipi.it

**Abstract:** The lack of financial resources and the constraints about interventions are threatening the survival of built heritage and the multiple benefits it can provide. In time, the role of building conservation has changed from preservation to being part of a sustainable strategy where adaptive reuse may allow to protect built heritage, while promoting it as a resource. This paper presents the results of a multicriteria analysis applied to the case study of Certosa di Pisa in Calci (Tuscany), a former Carthusian Monastery currently run as a publicly owned museum center. Based on information gathered from literature and the involvement of the two main stakeholders, a SWOT analysis was performed to identify three scenarios in which new functions were introduced with the aim to cover restoration and maintenance costs. Scenarios were compared by using a participatory MCA, taking into account not only economic performances but also cultural, territorial integration and restoration co-impacts. Results show that it is possible to reach economic sustainability while conserving heritage values, but several criticalities may hinder the process. Conclusions discuss the suitability of the method in identifying sustainable reuse solutions and highlight the role of governance bodies and the problems related to their public and/or private composition.

**Keywords:** adaptive reuse; built heritage; co-benefits; participatory multicriteria analysis; public-private partnership; listed buildings; religious buildings; Tuscany; Italy

## 1. Introduction

Italy is one of the world countries with the richest heritage assets, often consisting of publicly owned sites and buildings. These assets are not only valuable based on their market value, but mostly for the flow of cultural ecosystem services that they are able to provide. The uniqueness and irreproducibility of built heritage have a similarity with the natural capital, whose stock and structures can be threatened from over-use by human activities [1]. Costanza et al. [2], in a paper about ecosystem services, stress how, "*in order for these benefits to be realized, natural capital—which does not require human activity to build or maintain—must interact with other forms of capital that do require human agency to build and maintain. These include: (1) built or manufactured capital; (2) human capital; and (3) social or cultural capital*". Powel et al. [3] affirm that "*the cultural heritage values of buildings and structures can be incorporated into an ecosystem services framework, through considering them as both an integral part of their associated historic spaces and of their wider landscape settings*". Rizzo and Throsby affirm that "*regarding heritage as cultural capital invites consideration of sustainability aspects, in parallel with the treatment of natural capital in economic theory, allowing us to derive a sustainability rule for cultural capital accumulation*" [4].

The condition of man-made product implies that built capital cannot survive without adequate management actions. Consequently, while a stock of natural capital can benefit

from the absence of human activities, built heritage cannot survive without it. When built heritage is no longer suitable to perform its original functions either because these functions are no longer needed (functional or social obsolescence) or because the cost of providing them is no longer sustainable from an economic point of view (physical, technological, legal and economic obsolescence) [5], there is a high risk of abandonment and irreversible loss of it.

In Italy this risk is comparatively high due to the large endowment of built heritage, often in a condition of ruin and under-utilization due to the scarcity of public financial resources to face the costs of preservation and the lack of skills [6,7]. According to the Agenzia del Demanio (State Property Agency) statistics, in 2020 Italian State owned more than 5000 listed buildings [8] whose protection as public cultural heritage is ensured by the Ministry of Cultural Heritage through its territorial entities (Regional Secretariats and Superintendences for Architectural and Landscape Heritage) [6]. Napolitano and De Nisco point out that, although Italy ranked first on the "heritage & culture" dimension on the 2014–2015 edition of the Future Brand Country Brands Index, the economic contribution of the Italian cultural heritage sector was estimated at only 0.2 per cent of the country's GDP, although its indirect contribution as a component of the production process of heritage-related products and services is much higher [9]. This implies that, while heritage and culture are paramount for Italian economy, the direct revenue that they are able to generate is low and often unable to cover costs of their maintenance. As regards the use of public financial resources, from an analysis of Dalle Nogare and Galizzi, it emerges an "*electoral cycle in which the incumbent spends less on culture in an election year*". They "*interpreted these results in the light of the facts that voters in Italy may prefer other types of public expenditure to culture*" [10]. Cerquetti and Ferrara, in a recent analysis of the local cultural heritage perception among young generation, found out that "*the perception of cultural heritage as a common good is not well rooted*" and that "*physical and intellectual accessibility to cultural heritage was never recognized as a cultural right*" [11]. Consequently, the current situation of public debt and economic crisis could put pressure on politicians regarding public spending on cultural heritage, thus stressing the importance to ensure conservation through the enhancement of culture and heritage as an economic resource. As Della Spina states "*cultural heritage . . . can be considered not only a legacy to be handed down to posterity but also a central resource for triggering processes of local and global development*". Nevertheless, she points out how "*at the moment . . . the conservation and enhancement of the heritage that has a cultural and landscape value represent a burden for the community*" [7].

As Bullen and Love state for Australia, "*the role of building conservation has changed from preservation to being part of a broader strategy for urban regeneration and sustainability*" [12]. Nesticò et al. [13], stress how "*the European Framework Program for Research and Innovation (Horizon 2020) points out the positive effects that may result from the valorization of the public buildings of cultural heritage, as a synthesis of the traditional passive protection of these assets— that is proved unfit as well as financially unsustainable for the Public Administration—and their productive use, through modalities compatible with their nature and vocation*". According to Yazdani Mehr, "*in the contemporary era, adaptive reuse has been considered as a strategy for protecting these buildings for both present and future generations*" [14].

The problem of adaptive reuse of historic buildings has a multi-dimensional nature and presents a high complexity due to the intertwining of the aspects to be tackled. In some cases, the analysis has been performed by using a multi-criteria analysis (MCA) [15]. Indeed, the aspects involved span from the need to respect the architectural and artistic value of the buildings to the symbolic values they have for the local community up to the use as a resource for economic development and to the environmental implication of its restoration.

In this framework, we present a case of an adaptive reuse of a former Carthusian Monastery located in Tuscany, Italy, a Region well known worldwide for its outstanding artistic and cultural heritage and beautiful landscape. The case-study presents some peculiarity insofar as: a) it deals with a problem of a partial adaptive reuse that needs to

complement already existing uses; and b) poses specific problems of governance due to the hypothesis to include private for-profit activities in a complex where three public bodies are already involved, i.e., the Italian State, which owns the historic buildings, and Tuscany Region and University of Pisa, which manage the two Museums located in the building. More detail is given in Section 3.

## 2. State of the Art

As we have seen, trying to preserve a building that no longer has any function could cause its permanent loss, due to the lack of financial resources. According to Yazdani Mehr, "*the act of converting existing buildings to a new function in not new, since in the past, structurally sound buildings were changed to fit new functions or changed requirements, with little concern or questioning*" [14]. The same author provides a review and critical analysis of the principal 19th and 20th century theories of conservation and restoration and their implications for adaptive reuse of heritage buildings, since the adaptive reuse concept covers the concepts of conservation, restoration, preservation and even maintenance of heritage buildings. Nevertheless, she states that "*in the contemporary era, this is impossible to just focus on maintenance due to changes in user demands and advances in technology. These contemporary changes necessitate some levels of adaptation in heritage buildings especially for those which are still in use*" [14].

New functions may reconcile the conservation of specific heritage values with forms of management able to guarantee economic sustainability, i.e., to provide financial resources able at least to cover restoration and maintenance costs. Mısırlısoy and Günçe [16] affirm that preserved buildings should make their profits of the maintenance and rehabilitation works of the structures, since this is important for the future of built heritage. Yildirim [17] stresses the importance for action effectiveness to coordinate conservation plans with management plans, highlighting the relations between conservation and protection actions from the one hand, and management and enhancement actions, on the other. In other words, there is the need to find new functions for historic buildings, keeping them as a living and evolving part of the socio-economic system, able to ensure their permanence without negatively impacting on their heritage values.

According to Latham ([18], p. 12) "*Creative re-use allows us to save and protect our heritage, while exploring its value as a resource; it prompts us to re-interpret our architectural needs and cultural aspirations, and sparks originality of mind through the process of turning constraints into advantages*". Plevoets and Cleempoel provide a literature review on adaptive reuse as a strategy towards conservation of cultural heritage [19]. A list of reasons why adaptive reuse could be a good solution in the case of unused religious buildings is provided by Velthuis and Spennemann [20]. Although attitudes towards possible fates of built heritage at risk of abandonment and decay seems to be quite different between Dutch and Italian population, adaptive reuse seems to be a flexible solution in a wide range of societal situations. The importance of societal attitudes is stressed by Velthuis and Spennemann [20] who affirm that "*it is widely accepted that re-use often would not happen without a strong desire from within society to conserve and re-use a building*". From this point of view, the main difficulty is to find innovative solutions, able to reconcile social, economic and environmental points of view, that are considered positive by the entire society. To guarantee the collective interest in the process of enhancing cultural heritage, it is necessary to define sustainable strategies that must take conservation as a priority and, at the same time, be able to trigger virtuous circles of territorial and local development. These purposes may only be achieved by a public administration capable of governing the entire decision-making process that leads to a programming of the sustainable management of these assets, by equipping itself with tools to support decisions [7,13,21,22].

The problems related to adaptive reuse may vary according to the specific context. When financial resources are scarce and there are many buildings in need of intervention, it is useful to find a methodology for prioritization [23]. In the case that there is the need for a building fulfilling a specific function and more than a building that can do it after

reuse interventions, there is a need of a methodology able to choose the most suitable building [15]. Finally, in the case that the most suitable function needs to be identified for a specific building, there is a need of a methodology able to identify it [17,24–26]. Sometimes the above two problems may be faced simultaneously, trying to find at the same time the best combination of building and use among a set of available buildings and possible uses [7]. In some cases, it is possible for the building to maintain its original function, and in this case the reuse relates to the update of the way this function is fulfilled or to the introduction of sub-functions making the system more efficient.

The case study adds to the current literature insofar as it presents a situation in which some for-profit activities are introduced in underutilized or unused parts of an historic building in order to raise financial resources able to cover maintenance costs and possibly also restoration costs. The adaptive reuse of an entire building represents a simpler problem since there is neither the need to ensure "complementarity" to the existing activities and functions, nor to coordinate the needs and aims of several already existing public entities with new entities, maybe belonging to the private sector. Vice versa, in the case study presented in this paper, it was necessary to reconcile the positions of the Demanio dello Stato Agency (which owns the building) and the two distinct Museums that have in use the building and that should give up some of the space that they currently have in use, although not fundamental to their activities, in order to host new activities.

In accordance with the principle of sustainable protection of public real estate, the preventive evaluation of reuse choices has the aim of ensuring the safeguard of cultural values in the actions for enhancing existing buildings resources. In particular, the new functions must be able not only to protect the identity of the asset, guarantee significant growth in economic and social values, but also be feasible and sustainable in the long term from an economic point of view [21]. This evaluation is complex and multidimensional and needs to take into account a multiplicity of aspects. Literature analysis can help in building a general framework to be used as a check list of relevant criteria and attributes when addressing reuse issues [27]. Conejos et al. [28] propose a list of attributes describing Physical, Economic, Functional, Technological, Social, Legal and Political aspects. Wang and Zeng [26] organize their analysis by distinguishing Cultural, Economic, Architectural, Environmental, Social and Continuity aspects. Yildirim [17] highlights the need to consider characteristics related to Aesthetic, Spiritual, Social, Historical, Symbolic values and Authenticity when dealing with heritage buildings. Thus, the protection of public historic buildings may result in many differentiated co-benefits or co-impacts.

According to Mayrhofer and Gupta [29] in the last decades the term 'co-benefits' has become a predominant concept in scientific writings. IPCC ([30], p. 119) defines co-benefits as "*the positive effects that a policy or measure aimed at one objective might have on other objectives, without yet evaluating the net effect on overall social welfare*". IPCC specifies that "*Co-benefits are often subject to uncertainty and depend on, among others, local circumstances and implementation practices*". Mayrhofer and Gupta note that "*even if co-benefits are incorporated in policy design, they often face implementation challenges due to a lack of awareness, a guiding framework and common understanding among policy makers and bureaucrats*" [29]. They identify three strands of usage in empirical research, i.e., (a) Climate co-benefits for which the paradigm is "*Development first*" and there is a subordination of climate co-benefits, (b) Development co-benefits for which the paradigm is "*Climate first*" and there is a primacy of climate change and, (c) Climate and [other goal] co-impacts/co-benefits which are seeking synergies through an equal treatment of goals. While the co-benefits approach has mainly been related to environmental and economic goals, it is possible to make a parallel with cultural heritage policies. Thus when dealing with policies related to cultural heritage it is possible to find approaches related to economic development for which profit is the goal and cultural co-benefits arise only from the necessity to preserve heritage as a resource to get it; heritage preservation approaches with economic co-benefits, where the focus is in preserving cultural assets and economic benefits are related to the activities for ensuring its permanence; and more balanced approaches where there is the research for a synergy, keep-

ing in mind that cultural heritage could be a driving force for economic development and that its recover and preservation requires adequate economic and financial sustainability.

Co-benefits concept focuses on reconciling traditionally conflicting goals [29] and on identifying "*win-win*" solutions, where more than one goal may be pursued at the same time, rather than stressing trade-off among competitive goals, with the risk to deepen conflicts among stakeholder groups affected in different way by positive and negative impacts. However, according to Mayrhofer and Gupta [29], often the co-benefits concept "*ends up being a 'business-as-usual' incremental approach*", being a superficial addition to cost-benefit or cost-effectiveness analyses. Besides, they refer that "*several scholars caution that a search for 'win-win' options in line with the co-benefits approach obscures trade-offs*" and that "*there is a risk that the co-benefits concept is used to 'sell' particular policies in an opportunistic manner*". Hanson et al. [31], in a paper about nature-based solutions (NBSs), confirm that "*few empirical studies have implemented the idea of co-benefits, or systematically engaged with stakeholders for tasks other than being the empirical data sources, through interviews and surveys*", although they represent 'core ideas' of NBSs. Vice versa, the problem of adaptive reuse, as many other complex problems, asks for the involvement of researchers of different disciplines together with members of the society who have a stake in the specific problem, i.e., a multidisciplinary and transdisciplinary approach.

As stated above, the complexity of the problem is also related to the multiplicity of stakeholders, who usually have different views, sensitivity and priorities about objectives. They range to public government representatives to developers and owners, besides professionals such as architects and researchers like architectural historians [15]. Mısırlısoy and Günçe [16] affirm that, although the users' contribution in decision-making about adaptive reuse is very important, it is usually ignored in projects. In recent times, a new type of stakeholder is emerging, i.e., hybrid organizations resulting from long-term public-private partnership (PPPs) [21,32,33]. Participation of private entities may help to raise financial resources when public resources are not sufficient to ensure maintenance and management. PPPs may promote the creation of new jointly owned organizations, such as companies with public-private capital or cultural consortium, associations and foundations. In PPPs the public entity component is usually responsible for preservation-related activities, while enhancement related services are contracted out to private or not-for-profit enterprises [34]. Innovative and profitable uses of cultural heritage should include maintenance in the process of ordinary activities and productive uses. PPPs may represent an important and innovative tool for cultural heritage, albeit they may raise critical issues [35]. Despites problems and criticalities, Dubini et al. [34] describe experiences where the public-private cooperation was successful, although they stress the need of specific skills to manage the relationship.

In this context, this paper proposes an integrated evaluation model, which combines multi-criteria methodologies and economic-financial analyses, in order to prioritize alternative scenarios according to the expectations of the interested parties. Multi-stakeholder Decision Analysis aims to include multiple dimensions in the evaluation process to support the identification of reuse and sustainable development strategies, including knowledge of experts and the community [36]. In the present paper, special attention has been given to the two main stakeholders i.e., the managers of the two museums located in the historic building, in order to find for-profit activities able to supply financial resources without interfering with museum activities. The involvement of museum directors in the case-study analysis had also the aim to promote a positive attitude towards the new functions and to stimulate participation in the co-building of alternative scenarios. This resulted in some adaptations of the AHP method, as it would be better explained in the next section.

## 3. Materials and Methods

### 3.1. The Case-Study Area

As many other articles, this paper discusses the problem of heritage building reuse starting from a case study. The historic building chosen for the present analysis is the

Certosa di Pisa in Calci, a former Carthusian Monastery dating back to the 14th Century, which is located close to the city of Pisa, in the foothills of Monte Pisano. Certosa di Pisa is a state-owned listed building with a gross surface of about 18,500 square meters while the 'area di sedime' – i.e., the area obtained from the projection on the horizontal plane of the masonry and external load-bearing structures of the building located above the ground level and of the parts not covered by topsoil of suitable thickness - is about 5750 square meters [8]. A short description of the main features of the Monte Pisano environment and the Calci Municipality can be found in [37].

The Calci municipality belongs to the urban region of Pisa, which resulted from the joint collaboration of six municipalities that voluntarily decided in 2008 to have common urban planning tools [38]. Pisa is a university city located in the heart of the Tuscany region in Italy. The city has about 90,000 inhabitants and it is suffering from a serious demographic decline due to people moving to the nearby municipalities, especially families with children. Universities and tourism are the largest industries. However, tourism is seasonal and very few among the thousands of tourists who visit Pisa every year to see "*Piazza dei Miracoli*" and the famous leaning tower, spend the night in town, thus failing to have a broader impact on the local economy. Tourists are more attracted by the countryside surrounding Pisa, the famous campagna Toscana [39,40], that is rich of agritourism and high-quality local food [41].

The area, as the whole Tuscany, is characterized by very active capillary civic organizing and bottom-up initiatives [41]. Indeed, it was the "*Comitato Insieme per il Monte Pisano*" (Committee Together for the Monte Pisano) to mobilize the local community in 2014, after the detection of some problems threatening the state of the Monumental Complex of Certosa di Pisa and asking for urgent maintenance, to vote for Certosa in the Census promoted by Fondo per l'Ambiente Italiano (i.e., the Italian national trust fund) about "*luoghi del cuore*" (i.e., "*places of the heart*"), a competition aiming to distribute financial resources to the places that are more loved by the population. While this directly resulted in a relatively small contribute, the visibility gained favored a contribution of over 3,000,000 euros from the Ministry in charge of Cultural Heritage (MIBACT) [42]. Again, after a dreadful fire that burned down about 1,400 hectares on the area of Monte Pisano, with its natural beauty and heritage buildings, the same Committee promoted a mobilization getting 114,670 votes, the highest amount of ever since the competition started [43].

The choice to use Certosa di Pisa as a case study stems from two separate reasons. The first is that case-studies related to partial adaptive reuse and introduction of new functions that cannot be considered as ancillary—since they would use a significative part of the buildings and have the aim to reach economic sustainability—are, to our knowledge, almost absent. The second is related to the opportunity given by the University involvement, since it has financed a research project on Certosa di Calci conservation and enhancement involving several departments and providing knowledge about a multiplicity of research fields. This has allowed to gather information on technical and disciplinary aspects to be used within the multi-criterial analysis.

Certosa di Pisa in Calci is a former monastery, that was inhabited by Carthusian monks up to the beginning of the seventies. In 1978 its property was transferred to the state, and thereafter it was transformed into a museum center. It is located close to the municipality center of Calci and less than 15 km from Pisa. The Calci municipal administration is well aware of the problem of accessibility to Certosa and in February 2019 issued a notice of competition to gather new ideas on the improvement of the main road connecting its urban center and the Certosa. Graphic tables showing Certosa and Calci urban center locations and the main viability interconnecting them are available from the website of Calci municipality [44].

Figure 1 shows a view of the landscape in which Certosa di Pisa in Calci is located, while Figure 2 shows a detail of the main façade.

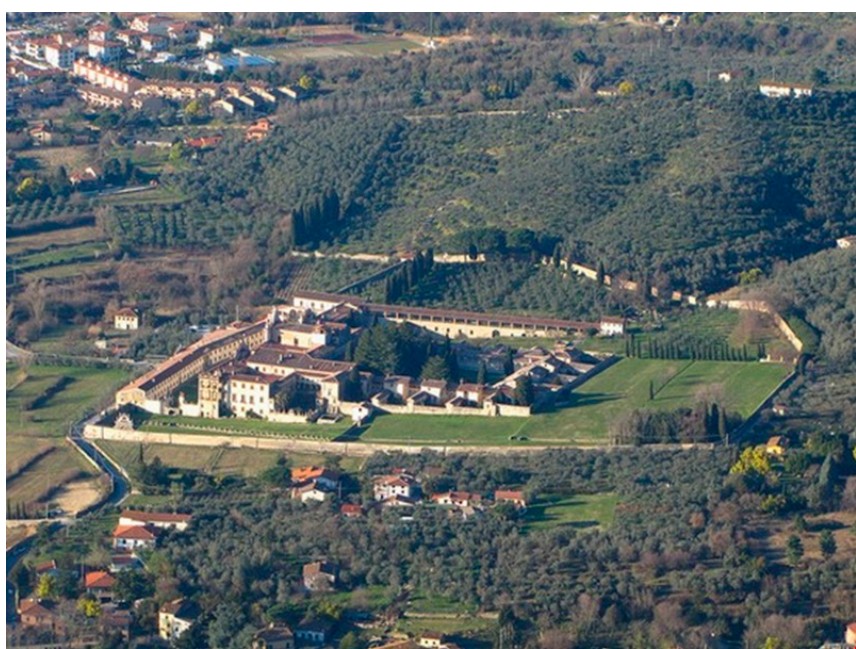

**Figure 1.** A view of the landscape where Certosa di Pisa in Calci is located (Certosa di Calci Pisa ©Michela Simoncini-CC BY 2.0 ©Michela Simoncini).

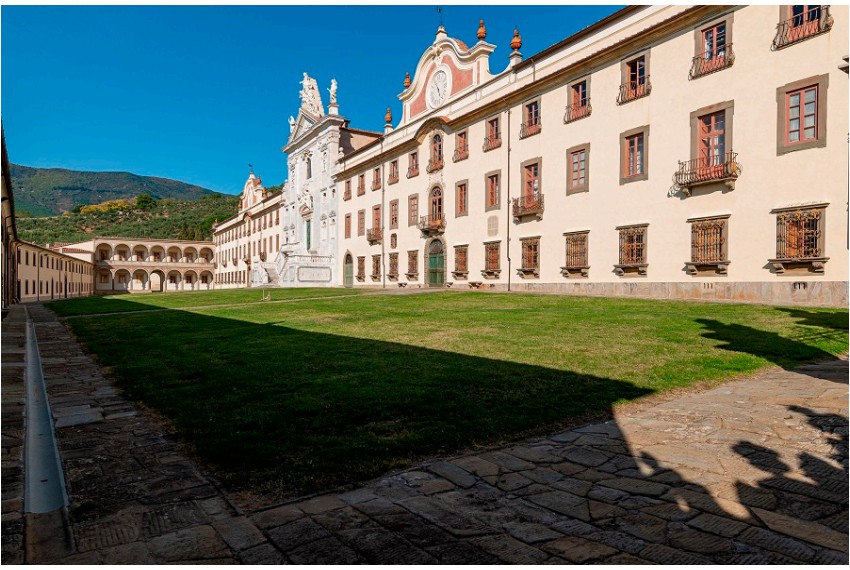

**Figure 2.** A view of the main façade of Certosa di Pisa in Calci (Source: University of Pisa website)

More pictures of Certosa and Monte Pisano can be found on the photo galleries of the 2014 and 2018 Census of Luoghi del Cuore, on the website of FAI [42,43].

Figure 3 presents a map of the site plan and the current use of spaces.

While the building still maintains its original layout, recent analyses highlighted that it is suffering from architectural and structural deterioration. At present the building hosts two distinct Museums, i.e., the National Museum of Monumental Certosa (NMC), belonging to the Tuscan Museum System and under the responsibility of the Tuscany Region, and the Natural History Museum (NHM), which belongs to the University of Pisa. The co-presence of two Museums under the responsibility of two distinct public entities makes the management quite difficult and hinders an adequate and coordinated market strategy. As a consequence, the complex is neither currently able to raise funds to cover annual costs of ordinary maintenance, that have been estimated to amount to

500,000 Euro/year [27], nor the cost of extraordinary interventions needed to solve the problems of architectural and structural deterioration.

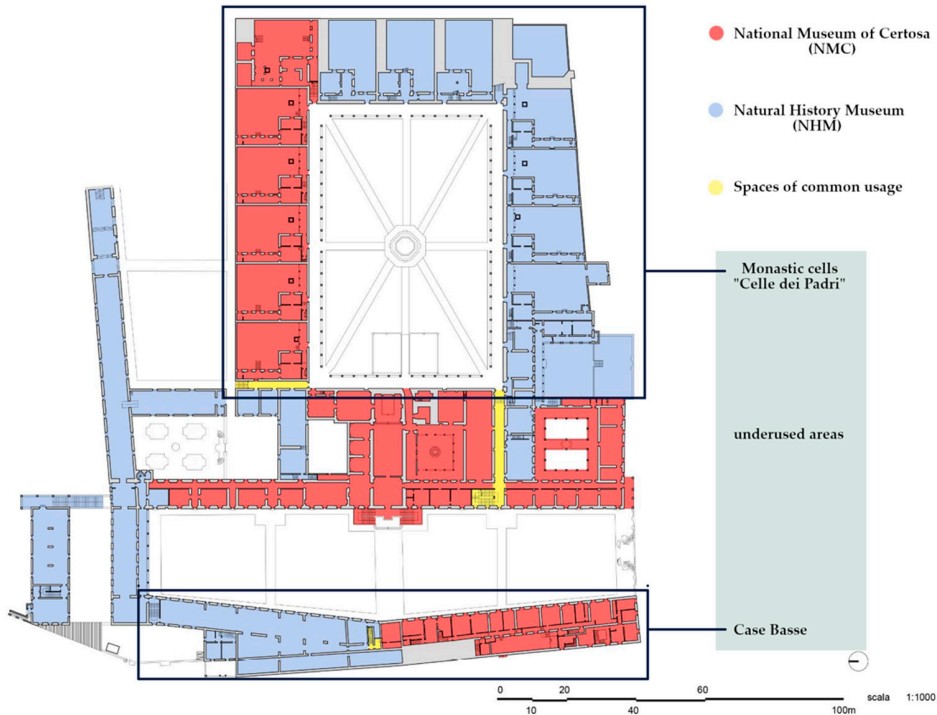

**Figure 3.** Certosa di Pisa in Calci: Site plan and current use of spaces (base on 1st level floor).

### 3.2. Aim of the Paper, Source of Information and Methodology for Scenario Identification

This paper provides an overall evaluation of alternative development scenarios, comparing different solutions related to the introduction of new functions and activities, with the aim to reach economic sustainability, seen as a necessary condition in order to ensure the conservation of this valuable historic building for future generations. The enhancement project has faced several criticalities. They were mostly related to the difficulties in coping with legal constraints and heavy bureaucracy when addressing interventions on listed historic buildings. Further criticalities were related to the problems in finding proper information on which to base alternative scenarios, to the need to experiment innovative ways of private-public cooperation in order to raise funds, and to the reluctance to change, not only from the part of officials working in the Agency in charge of cultural heritage, but sometimes also from the local population. Since a mere preservation is not economically viable, the only solution was to propose conservation actions able to respect as much as possible the heritage value of the building.

The scenarios were built on the basis of information coming from the following sources and activities:

(1) Studies and technical documents on the structural, historical, and cultural features of Certosa di Pisa.
(2) Review of the literature on the problems of adaptive reuse of heritage buildings, with a special focus on religious buildings, in order to identify relevant criteria and attributes for the specific case study.
(3) Review of the literature on museum marketing and management of publicly owned heritage buildings.
(4) Case-study analysis with the aim to identify best practices suitable to be adopted in the case of Certosa. This comparative analysis included 7 publicly owned museums (5 located in Italy and 2 in other EU countries) and 21 Carthusian Monasteries (16 located in Italy and 5 in other European countries).

More details about sources can be found in [45]. The above information has been used to build a SWOT analysis, shown in Table 1, in order to identify the best intervention strategies that were represented in three development scenarios.

**Table 1.** SWOT Analysis used to identify suitable enhancement scenarios [46].

| Strengths | Weaknesses |
|---|---|
| • Historic and artistic value<br>• Identity value<br>• Possible recovery of monastic traditions<br>• Significant naturalistic collections<br>• Educational offer differentiated by age<br>• Unused spaces available for new functions<br>• Wide open spaces | Unsatisfactory maintenance<br>Architectural and structural deterioration<br>Low security level<br>Unsuitable environment for disabled persons<br>Inadequate servicescape<br>Separate management for the two museums |
| **Opportunities** | **Threats** |
| • Cooperation with the University of Pisa<br>• Closeness to main infrastructures<br>• Closeness to attractor poles (Lucca and Pisa cities)<br>• Chance of getting financial resources from EU and private investors<br>• Availability of ICT to be used for promotion<br>• Connections with path networks<br>• Support by the municipality administration<br>• Incentive for the local economy | Unsatisfactory public transport<br>Absence of bike and pedestrian trails connecting to residential areas<br>Lack of parking places<br>Lack of local accommodations<br>Disconnection with the territory<br>Possibility to raise conflicts when involving local stakeholders in projects aiming to improve transport and connection infrastructures |

The analysis of best practices and previous similar experiences helped in drawing scenarios, e.g., in dimensioning spaces, locating services and functions, etc.

All scenarios had to meet the following requisites and characteristics:

(1) To identify new functions to be introduced in currently unused or underused spaces,
(2) To ensure economic sustainability by producing financial flows able to cover maintenance costs,
(3) To ensure that new functions are compatible with the heritage value of the building and with its already existing functions,
(4) To promote a synergy between public and private bodies/entities,
(5) To include a proper evaluation of economic and financial feasibility,
(6) To include other relevant attributes in order to take into account cultural and societal aspects and
(7) To involve professionals and stakeholders both in their definition and in the evaluation phases.

Three scenarios were individuated: i.e., Digital Detox, Education Facility and Coworking Spaces, which in this case constitute the alternatives among which to choose the best solution in terms of new functions to be introduced. Scenarios were evaluated by several attributes belonging to economic, cultural, territorial integration and restoration impact criteria. The hierarchical structure of the problem is described in Figure 4, while scenarios, criteria and attributes are described in detail in the following sub-sections.

### 3.3. Common Features to All Scenarios

The three scenarios had two features in common. First, the hypothesis to put the two existing museums under the umbrella of a single managing entity, under the legal form of a foundation, with the responsibility of the whole monumental complex. Then the creation and improvement of servicescapes, that has been deemed as a prerequisite for any form of development.

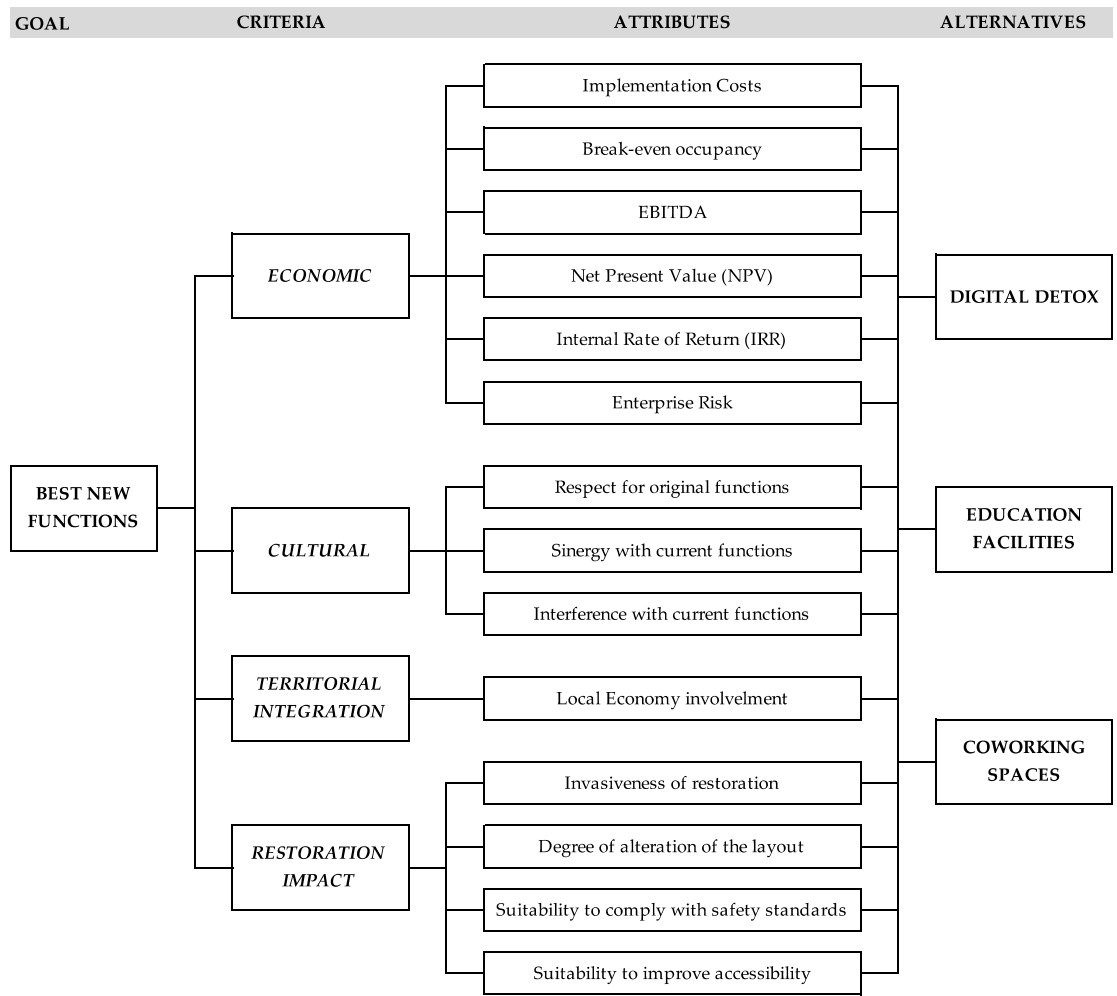

**Figure 4.** Hierarchical structure of the problem.

The choice of a foundation as a legal form for the managing entity rested on its eligibility to manage public heritage according to the Italian "*Cultural Heritage and Landscape Code*" (Legislative Decree 42/2004) and its possibility to have a public-private capital [35].

Servicescapes to be introduced consist in a ticket office at the service of both museums, a bookshop and other thematic shops selling local food products, herbal medicines and liqueurs made with aromatic and digestive herbs, typical products of a monastery, together with souvenirs related to the museums, a cloakroom, restrooms, a cafeteria and a restaurant. The restaurant should use local products and offer meals at a medium-high price level. The most suitable location for the above activities was deemed to be the ground floor of "*Case Basse*". Figure 5 presents the localization of servicescapes.

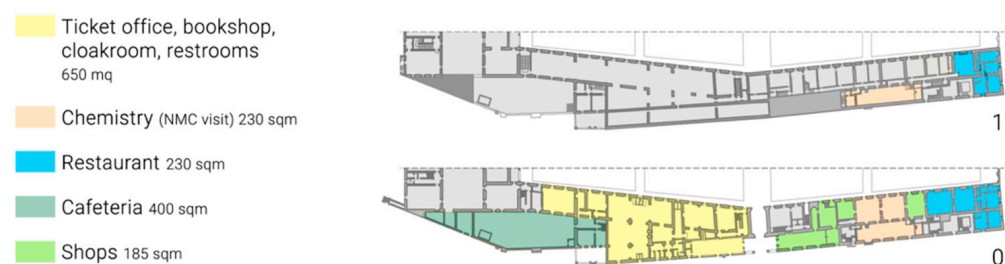

**Figure 5.** Preliminary intervention: servicescape localization. Number under the right side indicates floor level (Source: [27], modified).

Servicescapes may have multiple positive impacts, since they could attract private investors and enhance the experience of visitors, who may be induced to spend more time (and likely money) inside the monumental complex [47]. According to Dubini et al. [34], there is a risk in considering preservation-related activities as more important than enhancement-related activities, as usually do governmental officials, since "*from a visitor perspective the quality of the experience may be determined either by the splendour of the site visited or by the availability of a clean restroom*". Since most of servicescapes would be rented to privates, both a range of costs of restoration and income as monthly rent for the commercial spaces, have been estimated. More details are provided in Table 2.

**Table 2.** Costs and income of servicescapes (Source: [27], modified).

| COSTS | Min (€) | Max (€) | INCOME | Euro/Month | Euro/Year |
|---|---|---|---|---|---|
| Restoration and implementation of the new functions | 1,876,743 | 2,667,332 | Total rent of commercial spaces | 9325 | 111,900 |

The three scenarios that were hypothesized at the end of SWOT analysis aimed to enhance the current situation of Certosa di Pisa in Calci by creating either: (a) a digital detox accommodation structure; (b) a University complementary education structure providing accommodation for participants; or (c) Co-working spaces and labs for specialized craftsmen. The three scenarios are described in detail in the following sub-sections.

### 3.4. First Scenario: Digital Detox Accommodation Structure

The first scenario relates to the hypothesis to enhance the situation by introducing a Digital Detox accommodation in the space of monastic cells, except for two: one which would be part of NCM and another that would be used as reception [45]. The peace of an old monastery set in a beautiful environment is suitable to host this kind of activity, that is exploiting a specific tourist niche market. Modern life often promotes an overuse of technological devices, especially for high qualified workers, who often suffer for technostress, an occupational disease due to overuse of technological devices and, consequently, need detoxification periods.

The accommodation rooms would be set in 32 monastic cells, each destined to a single guest. In this case it is possible to maintain the original layout of the monastic cells, with only basic equipment, in order to help people to reconnect with their bodies and minds, and to avoid the creation of facilities asking for invasive restoration works. Other spaces of the first floor of "*Case Basse*" would be dedicated to activities offered by the accommodation structure and spaces were to experience a modern and laic interpretation of monks' life, trying to maintain a relation between old functions and new services. Among them, spaces for individual activities, as library and reading rooms (former anchoritic spaces), rooms for yoga, meditation, wellness and thematic retreats (former coenobitic spaces for collective ceremonials), and art and handcraft works (former productive spaces). The areas formerly used for pilgrim accommodation and refreshment would be transformed in a restaurant and cooking workshop area [27]. Figure 6 presents the spatial distribution of new functions for the scenario of Digital Detox.

### 3.5. Second Scenario: University Complementary Education Facility Providing Accommodation for Participants

In the second scenario the hypothesis to create a structure for university and professional short courses is analyzed. This in the spirit of Erasmus + EU program, that promotes continuous education, staff mobility and strategic cooperation and interactions. The target for this scenario is represented by undergraduate, masters and PhD students, university professors, freelance professionals and companies. Since the University of Pisa is already involved in Certosa di Calci because of the Natural History Museum, this scenario could represent an opportunity to widen the range of its educational offer, e.g., offering summer

and winter schools, masters and specialization training courses. The second museum (NCM) could be involved in teaching activities and their organization, if the topic is related to its activities. The existing classrooms, laboratories and conference room would represent useful resources for this scenario. Additional classrooms could be gained by converting five monastic cells, by setting them up with adequate furniture without any modification to the architectural layout. Workshops and activities would involve small groups, since safety standards increase in case of large groups using the same room, and this would highly increase costs. The remaining monastic cells and the former guesthouse of the monastery, situated on the first floor of "*Case Basse*", could be used for accommodating up to 52 participants. Figure 7 presents the spatial distribution of new functions for the scenario of Education facilities.

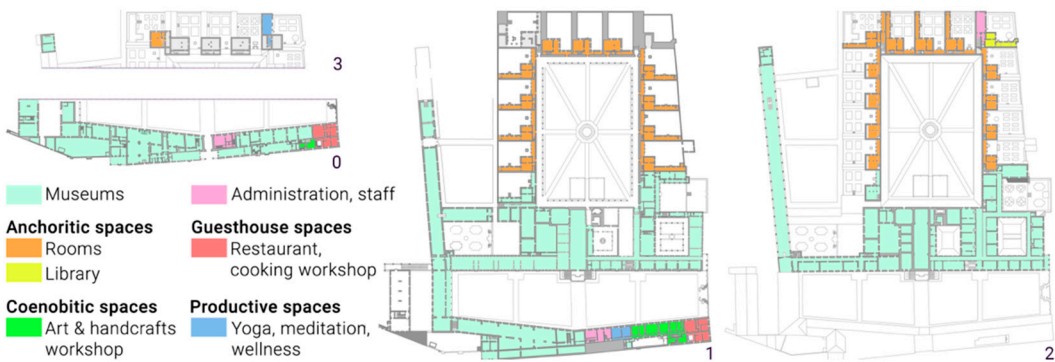

**Figure 6.** First scenario: Space distribution of new functions. Number under the right side indicates floor level (Source: [27]).

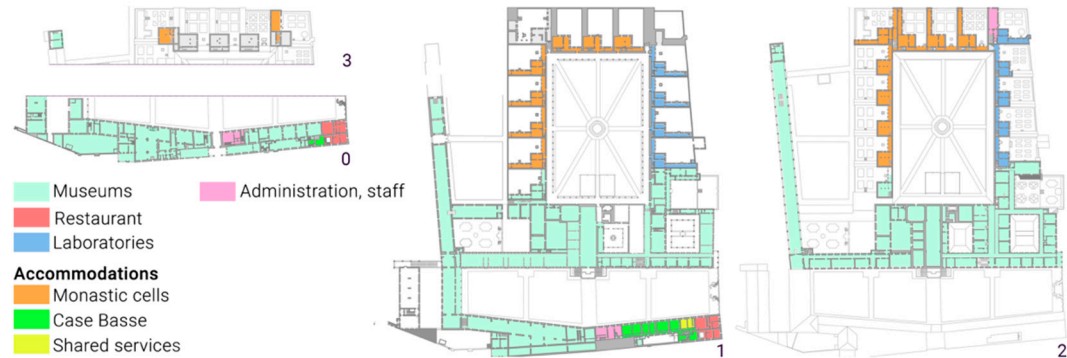

**Figure 7.** Second scenario: Space distribution of new functions. Number under the right side indicates floor level (Source: [27]).

### 3.6. Third Scenario: Co-Working Spaces and Labs for Specialized Craftsmen

In the third scenario the hypothesis is to transform unused spaces in a structure providing co-working facilities, spaces and services to local enterprises linked to cultural heritage and specialized crafts. The monastic cells and the "*Case Basse*" will be transformed into office spaces, both in private rooms and shared areas, and conference rooms for special events and workshops. The structure would provide services such as virtual office, consultancy, coaching, training, catering in the case of events or work meetings involving small groups, etc. External areas could be used for temporary exhibitions, to illustrate projects and offer services. In order to limit the invasiveness of restoration, working spaces would be partitioned by furniture and shared services would be localized only in some parts of the complex. Target for this scenario are small enterprises, freelance professionals, associations, start-up companies and spin-offs of the University of Pisa. Priority would be given to those companies developing, producing and providing services in the field of cultural heritage sustainability and tourism, especially if directly related to the activities of

the museum center. Thus, one of the aims is to involve the local economy, together with that to create a synergy among multidisciplinary skills required for the enhancement of cultural heritage albeit very seldom it is attainable by a small enterprise in its own. Figure 8 presents the spatial distribution of new functions for the co-working spaces scenario.

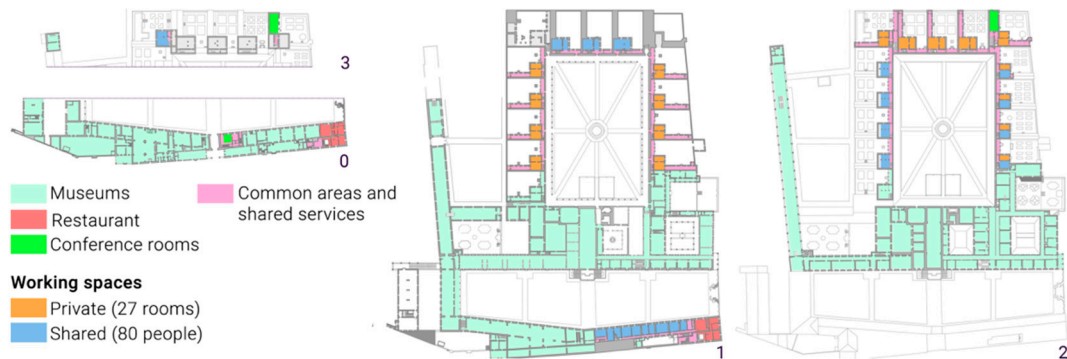

**Figure 8.** Third scenario: Space distribution of new functions. Number under the right side indicates floor level (Source: [27]).

### 3.7. The Analysis of Scenarios

#### 3.7.1. The Methodological Approach: MCA Participative Analysis

Due to the multi-dimensional nature and the complexity of the problem to be addressed, Multi-criteria analysis has been chosen as a suitable method to support decisions. This due to its capability to take into account both available information on relevant aspects and on stakeholders' values [15].

From the point of view of benefits, MCDM are neutrals, meaning that they are able to take into account a plurality of effects, independently from their nature. Thus, this approach may highlight trade-off between goals, the existence of co-benefits or "neutral" situations. From this point of view, the existence of co-benefits mostly rests on the previous part of the process, i.e., the one identifying innovative strategies and solutions, which should be able to overcome conflicting objectives and to propose solutions for overcoming conflicting situations.

Applied economists who have been working in the Environmental Impact Assessment field, cultural tourism development or physical planning have usually the capacity to interact with researchers, professionals and experts in other disciplines in their 'working toolkit' and this maybe may explain why, according to [29], the co-benefits concept "*has tended to attract initially natural science scholars and subsequently mostly economic scholars*" and "*This reflects not only the nature of the concept, which attracts economic analysis, but also the lack of engagement by other academic scholars who find it more difficult to develop theory on this concept because of their own specific epistemologies*".

An analysis of the relevant literature on multi-criteria decision-making (MCDM) methods in heritage buildings is provided in [48]. These authors highlight how, although MCDM methods are widely used in different fields, their application in the area of heritage buildings is still quite scarce. Within the analyzed literature, the three most used MCDM methods are AHP, ANP and fuzzy Delphi, although experts' evaluation is also popular. A comparative analysis of multi-criteria methods for the enhancement of historical buildings is provided by [49].

In this paper, in order to evaluate the three scenarios, a participatory multi-criteria analysis (MCA) was used. In this first explorative analysis, participation was limited to the main stakeholders, i.e., the directors of the two museums located in Certosa di Pisa. This for the awareness of the importance to involve directors in the adaptive reuse project and with the aim to better understand the limits of the current organization. Carbone et al. [50], in a paper about quality in the management of cultural heritage sites, shift the focus from the analysis of demand (visitor opinions) to the analysis of supply (manager

opinion) and highlight the role of 'cultural mangers'. Carbone et al. stress as "*in some cases, interviews revealed skepticism and even a sort of stigmatization on the part of cultural heritage managers towards the idea of quality. This attitude has proved to be common in those participants who admitted not to feel comfortable in talking about 'management'/'quality management' when referring to cultural heritage. In their opinion, those concepts lead back directly to the economic, for-profit commercial sector, thus distant from the mission and the values of the cultural sector, even antithetical*" [50]. Vice versa the same authors stress how cultural managers should be "*engaged in achieving a balance between developing the tourism industry, generating revenue while still conserving the authenticity of intangible heritage as well as the physical integrity of heritage sites, objects and collections (tangible heritage), promoting and celebrating their educational, historic and cultural values*".

3.7.2. Criteria and Attributes

Attributes, on the base of literature analysis and its relevance in the specific case study, were classified into the following criteria: economic, cultural, territorial integration and, restoration impact. The attributes belonging to each criterion are illustrated in Figure 4 and in Table 3a–d [27]. Quantitative economic attributes have been chosen among those commonly used in financial and economic analyses. The other attributes were introduced to take into account qualitative aspects (co-benefits or co-impacts) that are deemed to be important although they cannot be easily monetized. This with the aim to have a holistic approach to the problem, rather than simply introducing qualitative ancillary remarks to a cost-benefits or cost-effectiveness analysis.

Economic characteristics for each scenario (included the common interventions related to servicescape) were computed on the base of technical information and prices taken from different documental sources, i.e., the "*Prezziario Restauro dei beni artistici*" edited by Genio Civile (the Italian Civil Engineering Office) in 2019 [51], that lists unitary costs of elementary actions for the restoration of art heritage and a similar handbook edited in 2018 by the Tuscany Region for Pisa province that lists unitary costs to be applied in the case of public works. For most of the economic and financial aspects it was possible to compute quantitative attributes, based on an estimate of flows of income and costs.

Table 4 shows the value of attributes belonging to the economic criterion, which were calculated according to the methodology presented in [45]. The three scenarios have similar costs since restoration works are mainly aiming to the conservation of the original layout, taking into account only low impact solutions. Since restoration costs are much higher than ordinary costs, also total costs are similar for all scenarios. Main differences in costs are accounted by furniture, equipment and the need to partition spaces in order to improve functionality. Restoration costs are suffering for higher uncertainty since some costs would be properly assessed only at the stage of design project. There is the risk for costs to be higher in the case that some unforeseen structural or other problems arise during the restoration process, as it often happens when restoring very old buildings.

It could be interesting to note how restoration works may have same ancillary co-benefits that in the present analysis have not been taken into account. According to Borri and Corradi [52], "*heavy structural-oriented interventions as well as the underestimation of the importance of the structural safety for masonry monuments has produced, in the past, irreversible damage to important buildings and monuments, and loss of architectural heritage in Italy*". Nevertheless, nowadays "*the application of retrofitting strategies, while improving the seismic performance of historic . . . buildings, will not significantly alter their appearance, will be reversible, and fall within the principle of minimum intervention*" [52]. Interventions related to safety standards (e.g., fire safety rules, requirement on seismic design, etc.) could have as a result not only a higher security for people working or visiting the site, but also the effect to protect historic buildings from risks of fire or seismic events.

**Table 3.** (**a**) Attributes belonging to the Economic criterion. (**b**) Attributes belonging to the Cultural criterion. (**c**) Attributes belonging to the Territorial integration criterion. (**d**) Attributes belonging to the Restoration impact criterion.

| a. Attributes belonging to the Economic criterion | |
| --- | --- |
| Attribute | Attribute value/score |
| Implementation (construction) costs | Restoration costs and costs for actions needed in order to implement the new functions characterizing each scenario |
| Break-even occupancy ratio | Occupancy minimum rate able to cover yearly maintenance costs for the whole "Certosa" |
| Earnings before interests, taxes, depreciation and amortization (EBITDA) | Profit from selling goods or services before costs not directly related to producing them |
| Net Present Value (NPV) | Net Present Value |
| Internal Rate of Return (IRR) | Internal Rate of Return |
| Enterprise Risk | Risk assessment of required investments, based on their capacity to generate profits |
| **b. Attributes belonging to the Cultural criterion** | |
| Attribute | Attribute value/score |
| Respect for the original functions | How and how much the new proposed functions take into account the fact that the building was originally a Carthusian monastery |
| Synergy with current functions | How and how much the new proposed functions may promote the already existing museal functions of "Certosa" |
| Interference with current functions | How and how much the new proposed functions may interfere with the flows of persons and with the museum activities which already exist |
| **c. Attributes belonging to the Territorial integration criterion** | |
| Attribute | Attribute value/score |
| Local economy involvement | Assessment of the capacity to integrate local economic activities and the new proposed functions |
| **d. Attributes belonging to the Restoration impact criterion** | |
| Attribute | Attribute value/score |
| Invasiveness of restoration | (a) Preference for introducing the new proposed functions, although they ask for major structural and plant interventions vs. (b) Preference for preserving the original building characteristics, although it implies the impossibility to introduce the proposed new functions or the necessity to modify them |
| Degree of alteration of the (spatial) layout | Assessment of the extent with which the new proposed functions would modify the original (spatial) layout of the building (e.g., by using space-dividing furniture) |
| Suitability to comply with safety Standards | (a) Preference for introducing the new proposed functions, although they ask for more invasive actions in order to comply with safety standards vs. (b) Preference for preserving the original building characteristics, although it implies the impossibility to introduce the new proposed functions or the necessity to modify them |
| Suitability to improve accessibility | (a) Preference for introducing the new proposed functions, although they imply the need for all spaces to comply with accessibility rules and, consequently, to look for alternative solutions or to undergo major works in order to guarantee accessibility vs. (b) Preference for preserving the original building characteristics, preferring functions that do not require accessibility for all spaces |

**Table 4.** Quantitative original values of attributes belonging to the economic criterion (source: [27]).

| Attributes Belonging to Economic Criterion | Scenario 1 Digital Detox | Scenario 2 Education Facilities | Scenario 3 Coworking |
|---|---|---|---|
| Implementation Costs | 7,121,506€ | 6,598,331€ | 6,835,474€ |
| Breakeven occupancy ratio | 64% | 80% | 60–61% |
| EBITDA | 146,648€ | 109,376€ | 108,374€ |
| NPV | 995,912€ | 579,546€ | −214,839€ |
| IRR | 12.17% | 9.82% | 2.99% |
| Enterprise Risk | High | Low | Medium |

The best alternative when considering only financial parameters is Digital Detox, but it is also the one showing the highest enterprise risk. For attributes which were not belonging to the economic criterion, only qualitative ordinal scores where initially given in a three steps scale. All qualitative scores have been transformed on numeric values based on the opinion of the directors of the two museums, considered as key stakeholders in the process of evaluation. Directors were interviewed in separate sessions in order to prevent mutual influences. However, their opinions were similar in most cases. Thus, the synthetic judgements assigned to each attribute by both the NHM and NMC directors were converted to a single score using the average value. Attribute scores were then normalized in a 0 to 3 scale and then multiplied to the corresponding weights (described in the following sub-section) both for qualitative and quantitative attributes. The final evaluation matrix including weighted normalized scores is presented in the Results section. In the case of territorial integration, e.g., scores given were 1, 2 and 3, while the weight assigned to this attribute was 0.100; thus, as a result, scenario 1, 2 and 3 had a weighted normalized score respectively of 0.100, 0.200 and 0.300. The use of a "*semantic scale*" with only three steps, rather than the usual 9 steps used in AHP, made it easier for stakeholders to express their opinions, that in this way were more reliable, although less precise.

### 3.7.3. Weights of Criteria and Attributes

The weights used for weighting criteria and attributes, were elicited from key stakeholders, i.e., museum directors, through a pairwise comparison [53]. A weight was assigned to each criterion in comparison with other criteria and the same process was applied to attributes. The sum of weights of all attributes belonging to a criterion coincide with the criterion weight. The sum of weights of all criteria was normalized to 1. In this way each attribute had a priority depending both on the importance of the criterion to which it belongs and on the importance of the attribute itself among all the attributes belonging to the same criterion. According to directors' opinion, who seemed to share the same perception about criteria importance, the most important criterion was the cultural (weight 0.400), followed by the economic (weight 0.300) and restoration impact (weight 0.200). Territorial integration was deemed to be the least important criterion (weight 0.100). Weights given to attributes and criteria are shown in Table 5. It is important to note that weights were expressed in relation to the specific problem under analysis and the proposed solutions. This means that the comparatively low weight given to the restoration impact may well have been due to the fact that in building scenarios only comparatively low impact solutions had been considered. At the same time, the low importance given to territorial integration is probably due to the fact that the two museum directors do not feel involved in the problem of the territory in which Certosa is located and on the spillover effects that could result from Certosa enhancement. Vice versa, from an interview of a Calci municipality administrator emerged both the importance of Certosa as an identity value and the scarce integration of the monastery with the surrounding territory, as witnessed by the fact that tourists interested in naturalistic or sportive activities in Calci area do not visit the Certosa and vice versa [46].

**Table 5.** Weights of criteria and attributes (Source: [27]).

| Criterion | Criterion Weight | Attribute | Cumulative Weight |
|---|---|---|---|
| Economic | 0.300 | Implementation (construction) costs | 0.029 |
| | | Breakeven occupancy ratio | 0.014 |
| | | EBITDA | 0.043 |
| | | NPV | 0.079 |
| | | IRR | 0.064 |
| | | Enterprise Risk | 0.071 |
| Cultural | 0.400 | Respect for the original functions | 0.200 |
| | | Synergy with current functions | 0.133 |
| | | Interference with current functions | 0.067 |
| Territorial integration | 0.100 | Local economy involvement | 0.100 |
| Restoration impact | 0.200 | Invasiveness of restoration | 0.060 |
| | | Degree of alteration of the (spatial) layout | 0.040 |
| | | Suitability to comply with safety standard | 0.080 |
| | | Suitability to improve accessibility | 0.020 |

## 4. Results

This section presents both the results of the evaluation analysis, which have been quantified via scores and weights and some more comments that have emerged in the interviews of the two directors of NHM and NMC museums and that were not introduced into the multi-criteria analysis.

### 4.1. Evaluation Results

As a result of the process described in Section 3, it was possible to build a table of weighted normalized scores for each attribute and scenario, were the total score for each scenario express an overall evaluation of the scenario performance in terms of economic, cultural, territorial integration and restoration impact. Weighted normalized scores are presented in Table 6.

The scenario that presents the best overall score is that related to a structure providing university complementary education and accommodation facilities for participants. Indeed, the chance to build and strengthen previous relationship with the University of Pisa reduces the enterprise risk and makes synergies with current functions easier, since museum related research activities could be enhanced by the help of the University. Hosting small groups allows to keep lower safety standards for avoiding the need for more invasive restoration actions and it is suitable for the use of Certosa spaces, which are not very large [27]. According to the present analysis this scenario is to be preferred over the other two, which show a similar and significantly lower overall evaluation score, although for different reasons. While this scenario seems to present features of a win-win solution, showing for all criteria a criterion total score that is the highest or close to the highest, Digital Detox and Coworking scenarios shows significant trade-offs among criteria, although they have a similar overall score, insofar as Digital Detox has significantly better performance in economic and cultural criteria, while Coworking shows better performance in territorial integration and restoration impact criteria.

### 4.2. Further Observations by Stakeholders

From the interviews made to stakeholders in order to elicit scores and weights for attributes and criteria, emerged also qualitative remarks that, while difficult to be introduced in the MCA process, could add further details to the evaluation. The main points have been listed below.

**Table 6.** Weighted normalized scores for attributes and overall scores for the three scenarios (Source: [27], modified).

| Criterion | Attribute Scenario | Digital Detox | Education Facilities | Coworking Spaces |
|---|---|---|---|---|
| Economic | Implementation (construction) costs | 0.029 | 0.086 | 0.057 |
| | Breakeven occupancy ratio | 0.043 | 0.014 | 0.029 |
| | EBITDA | 0.129 | 0.064 | 0.064 |
| | NPV | 0.236 | 0.157 | 0.079 |
| | IRR | 0.193 | 0.129 | 0.064 |
| | Enterprise Risk | 0.071 | 0.214 | 0.143 |
| | Total score for Economic criterion | 0.701 | 0.664 | 0.436 |
| Cultural | Respect for the original functions | 0.600 | 0.400 | 0.200 |
| | Synergy with current functions | 0.133 | 0.400 | 0.267 |
| | Interference with current functions | 0.067 | 0.167 | 0.167 |
| | Total score for Cultural criterion | 0.800 | 0.967 | 0.634 |
| Territorial integration | Local economy involvement | 0.100 | 0.200 | 0.300 |
| | Total score for Territorial integration criterion | 0.100 | 0.200 | 0.300 |
| Restoration impact | Invasiveness of restoration | 0.060 | 0.150 | 0.150 |
| | Degree of alteration of the (spatial) layout | 0.040 | 0.100 | 0.100 |
| | Suitability to comply with safety standard | 0.080 | 0.200 | 0.200 |
| | Suitability to improve accessibility | 0.020 | 0.050 | 0.050 |
| | Total score for Restoration impact criterion | 0.200 | 0.500 | 0.500 |
| Overall evaluation | Total score | 1.800 | 2.331 | 1.869 |

According to stakeholders, a further attribute that could have been introduced in the analysis is the degree of flexibility in the use of spaces. While scenario 1 is characterized by a fixed layout, in scenarios 2 and 3 the partition is mostly made by furniture that could be easily rearranged. In particular, it would be possible to organize exhibitions of different kind that ask for different space and layout only by rearranging furniture or its position. This would make it possible to shift from exhibitions presenting the results of workshops related to museums' activities to exhibitions promoting products of private firms, and consequently optimize the use of spaces in time and enhance synergies between pre-existing functions and local economy.

In scenarios 1 and 3, the NCM keeps the use of one of the monastic cells with a problem of interference between the flux of NCM visitors and the flux of users introduced by the new functions of these scenarios. This issue makes scenarios 1 and 3 less favorable than scenario 2, for which all monastic cells could be converted to laboratories, while guests would be hosted only in the "*Case Basse*" area. This implies a change of the hypotheses made in the present analysis and would ask for a new feasibility analysis for the modified scenario 2. A further solution could be that to host guests in accommodations near the Certosa, with a reduction of the sharing experience of participants, but improving the level of the territorial integration by involving external actors of the tourism sector. For scenarios 2 and 3 it would be possible to reduce the interference with current functions by concentrating their activities only in specific times of the year, but this would reduce their rate of occupancy and, consequently, their economic sustainability.

Stakeholders expressed the idea to introduce a further scenario as a result of a modification of scenario 1 aiming to create an accommodation structure where it would be possible to recreate in a more faithful way the old life of Carthusian monks by introducing

seclusion and the silence rule. Target for this specific accommodation offer would be people doing knowledge work at high level, who need programmed retirement to perform their tasks, without any external interference. While the risk in implementing this variant of scenario 1 would be higher than that of the original one, it would reduce the amount of costs needed for restoration, due to the fact that only one guest would be hosted in each cell and this would consent to keep the intervention as conservative as possible and at the same time to stay inside the compulsory safety standards. Besides, having less guests and almost always staying inside their cells would reduce the problem of interference with the flow of museum users, although there is the risk that visitors to the museums would disturb guests in seclusion.

## 5. Discussion

The participative method discussed in the present paper, although representing a preliminary analysis to be broadened by involving further stakeholders and deepened by analyzing in more details the scenario more suitable to positive development, contribute to the process of decision making in the case of cultural heritage for which there is a need to find a proper reuse in order not to risk a definitive loss. The comparison with best practices and other already existing experiences has shown that it is possible to introduce new functions in historic buildings without a loss of their architectural, artistic, cultural and identity values and at the same time to raise financial resources able to ensure a proper maintenance. As in many other cases, it is a learning process where the analysis of results may introduce new ideas about relevant attributes and scenarios that may improve the final results of the analysis. It is the case of the suggestions expressed by the two directors of Certosa complex museums who, at the end of the participatory process, were more involved and willing to give their contribution.

From the evaluation matrix that represents the results of the analysis at the current stage, it emerges as one of the scenarios under hypothesis seems to have the characteristics for being a win-win solution, were all the criteria have a good score, if not the best one. However, the suggestions coming from stakeholders, the new context determined by Covid 19 pandemic and the risk related to this kind of intervention ask for a prosecution of this evaluation process in order to take a proper decision. The risk is related to several aspects that could be summarized as follows:

- Risk of higher restoration costs, as often happens with restoration of historic buildings, although the level of detail with which they have been assessed during the project is quite good in comparison with other contributions.
- Risk related to management. Each of the proposed scenarios requires the presence of a high qualified management, able to organize functions and activities at their best in order to attract users in a suitable way and in a proper number. This is in line with the findings of [50].
- Risk related to the creation of a foundation with public-private capital. This new management entity should coordinate not only ideas, management and visions of the two public museums already operating inside the monumental complex of the Certosa di Pisa in Calci, but also be able to harmonize visions and goals of public sector officials with those of subjects of the private sector. Although, how stated in the "State of the art" section, there are some positive experiences about the role of public-private partnerships in the cultural heritage fields, this does not mean that to build a fruitful cooperation would be possible or easy also in this case.
- Risk related to the market response to the new functions/services that characterize each scenario. From this point of view, all the analysis needs to be updated to the new situation in tourism and educational markets determined by Covid 19 pandemic.
- Risk related to political scenarios. As we have seen, some authors found a negative relation between political instability and willingness to invest in the cultural sector. Besides, using cultural heritage as a driving force for development while maintaining its values would require clear policy goals and rules to be maintained in times, since

these are strategies that need a medium—long term horizon to be successfully implemented. Agenzia del Demanio has recently shifted from a strategy of dismission of state-owned buildings to a strategy of valorization by giving them in use to private entities for a long term under the condition of their development [6]. Nevertheless, due to political instability, the Certosa di Pisa project is currently in "*stand-by*".

As regards the potential of the co-benefits approach, this rests in our opinion mostly in a broader "*political vision*" where stakeholders involved in different disciplinary fields and with different stakes abandon an approach of pure defense of their opinion and short-term interests and try to 'leapfrog' to an approach promoting not a simple compromise but active synergies among positive effects of policies and measures.

In the fields of heritage buildings adaptive reuse, we think that a policy trying to promote a conservative enhancement rather than pure preservation could, in the long run, better contribute to the maintenance of our rich endowment of historic buildings and at the same time produce multiple co-benefits in terms of local development, identity building, etc. This does not mean that trade-offs should be ignored, as well as the risks related to the involvement of private stakeholders in the management of cultural heritage. In other words, in our opinion, a proper approach would be not the one to ignore existing trade-off but to actively seek for innovative solutions able to overcome them, as much as it is possible. As stated by Latham ([18], pp. 12–13), regarding building reuse: "*The real limitations are not archaeological, aesthetic, economical or functional, but psychological: the limits created by preconceptions, and by lack of imagination. Once the will is there, the skill and ingenuity will follow*".

**Author Contributions:** Conceptualization, M.R. and M.A.; methodology, M.R.; validation, M.R. and A.A.; formal analysis, A.A.; investigation, A.A.; resources, M.R.; data curation, A.A; writing—original draft preparation, A.A., M.R. and M.A.; writing—review and editing, M.A.; supervision, M.R.; project administration, M.R.; funding acquisition, M.R.; All authors have read and agreed to the published version of the manuscript.

**Funding:** This research was funded by University of Pisa. Title: Studi conoscitivi e ricerche per la conservazione e la valorizzazione del Complesso della Certosa di Calci e dei suoi Poli Museali. (Studies and researches for the conservation and enhancement of the Certosa di Calci complex and its museums).

**Institutional Review Board Statement:** Not applicable.

**Informed Consent Statement:** Not applicable.

**Data Availability Statement:** Data sharing not applicable.

**Acknowledgments:** The authors want to thank the anonymous reviewers for their comments that were very useful in improving the first version of this paper.

**Conflicts of Interest:** The authors declare no conflict of interest.

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
