# Peer review of "Adaptive Reuse of a Historic Building by Introducing New Functions: A Scenario Evaluation Based on Participatory MCA Applied to a Former Carthusian Monastery in Tuscany, Italy"

_sustainability, doi:10.3390/su13042335_

Round 1

Reviewer 1 Report

It seems that the accessibility to the Carthusian Monastery does not affect on the difference scenarios while it is important to understand how the Monastery is accessibile from the town and the road access to this Heritage.

Author Response

We have included in the paper some information about the location of the Carthusian monastery. It is quite close both to Calci Municipality urban centre and Pisa city centre. Although the currently existing bus service (the bus stop is at about 4 minutes on foot by Certosa) should be improved, probably accessibility is only a secondary aspect in differentiating scenarios, since all of them do not hypothesize high flow of people. As stated in the paper, Calci municipality is currently involved in the improvement of the road from Calci to the Certosa.

Reviewer 2 Report

The argument of the manuscript is very interesting, since adaptive reuse, in particular regarding Monuments and Goods of Cultural Heritage, is an important and very current topic. The adaptive reuse of Monuments like monasteries, castles or palaces, is indeed the best way to enhance them and consequently to preserve them in a planned and continuous manner.

The work deals with the theme in a complete and in-depth manner. In fact, through the analysis of a particular case-study, the authors carry out a careful and complete analysis of all the elements that concur to identify a valid potential solution.

The argument is well explained and theoretical methods are clearly described.

In the opinion of this referee, the paper can be accepted for publication, after taking care of some editorial revisions:

  • Paragraph 2.6 is repeated twice: review the numbering of paragraphs
  • Figures, with the exception of Figure n.1, are actually tables, even if data are not always numerical: why do the authors consider them figures? Is there a technical reason?

Author Response

We have taken care of the “double” 2.6 sub-section

We have transformed all relevant figures in tables. The reason for which they were put as figure was that we had previous bad experience about the way table have been formatted in the processing of a contribution.

Reviewer 3 Report

"The Evaluation of Alternative Scenarios based on Participatory Multicriteria Analysis for a Cultural Heritage Enhancement Project. The case study of Carthusian Monastery of the “Certosa di Pisa” in Calci (Tuscany – Italy)"

1. Introducción:

The contribution of this paper is limited by its current focus and structure. The authors need to properly centre the discussion and contribution of the paper within the current literature. What this paper can add to the cultural heritage literature is not clear. The authors need to properly centre the contribution with the data and the analysis. This contribution and focus should be present in the title. The title is too long, too general. Consider the progression of the argument in the introduction.

Describe knowledge gaps. Make clear in which tradition you position your study and the debate you like to contribute to.

P2-L53 "Thus, the lack of an adequate flow of financial resources could bring about the loss of a capital that has very significant option or existence values. Economic development has often been  considered as a conflictual goal with the preservation of built heritage, with the conse- quent need to analyze trade-off between economic benefits and cultural benefits". Why?? More explanations are necessary

Please clearly differentiate the introduction section from a new literature review section.The literature review should be expanded to include more details in all areas. There are several cultural herritage that I can recommend:
(2014) Towards a community-based cultural heritage resources management (COBACHREM) model or (2018) The contribution of cultural events to the formation of the cognitive and affective images of a tourist destination.

2. Materials and Methods:

(P4-L172). Needs further elaboration. Sampling criteria are nor clear, how where the study area selected? Is the sample representative?

Figure 1. Certosa di Pisa in Calci: Site plan and current use of spaces: Copyright??

Figures 3 and 4 have poor quality, it looks bad (Figure 3. Table of criteria and attributes; Figure 4. Table of weights of criteria and attributes)

3. Results:

Is there a relationship between the literature review and the results section of this paper? 

Are there any discrepancies on result?

4. Discussion

Very limited findings and discussion section relating to the literature. The implications are weak and I miss the conclusion section about cultural heritage and social impacts.

Could more text on the findings be added? The managerial implications need to be less routine and more interesting and relevant to your findings.

5. Others

The language could be improved. Generally it is OK but often lacking “flow" and abrupt

Author Response

We tried to better focus the paper and to give a clearer idea of the content by changing the title

We introduced a separate “state of art” section after “introduction” section. Given the aim of the paper, the literature review focusses on: a) adaptive reuse; b) co-benefits; c) public-private partnerships

We provided some insights on the reason why the case-study has been selected and how it adds to available literature

We have taken care of source and copyright for figures related to Certosa.

We have split former table/figure 3 in a figure illustrating the hierarchy of the problem (goal-criteria-attributes-alternatives) and in four tables, one for each criterion, trying to improve not only their quality in terms of readability but also the presentation of the research approach

We have integrated several parts of the text.

We went through English language.

Round 2

Reviewer 3 Report

Dear Authors:

I thank the authors for this revision work "The Evaluation of Alternative Scenarios based on Participatory Multicriteria Analysis for a Cultural Heritage Enhancement Project. The case study of Carthusian Monastery of the “Certosa di Pisa” in Calci (Tuscany – Italy)", which satisfies most of my concerns about the paper 

Originality:Now OK.

Introducction: Now it is correct

Materials and methods:Now OK.

Results:Now OK.

Discussion:Now OK.

Conclusions/ Implications: Ok, but suggestions for future research: additional managerial implications in line with the findings of the  study "The Evaluation of Alternative Scenarios based on Participatory Multicriteria Analysis for a Cultural Heritage Enhancement Project. The case study of Carthusian Monastery of the “Certosa di Pisa” in Calci (Tuscany – Italy)".